# Moderate Soil Drying-Induced Alternative Splicing Provides a Potential Novel Approach for the Regulation of Grain Filling in Rice Inferior Spikelets

**DOI:** 10.3390/ijms23147770

**Published:** 2022-07-14

**Authors:** Zhenning Teng, Qin Zheng, Bohan Liu, Shuan Meng, Jianhua Zhang, Nenghui Ye

**Affiliations:** 1College of Agronomy, Hunan Agricultural University, Changsha 410128, China; sailingtzn@163.com (Z.T.); zhengqin@stu.hunau.edu.cn (Q.Z.); liubohan@hunau.edu.cn (B.L.); mengshuanz@outlook.com (S.M.); 2Shenzhen Research Institute, The Chinese University of Hong Kong, Shenzhen 518057, China; 3Key Laboratory of Crop Physiology and Molecular Biology, Ministry of Education, Hunan Agricultural University, Changsha 410128, China; 4Department of Biology, Hong Kong Baptist University, Kowloon, Hong Kong 999077, China

**Keywords:** alternative splicing, microRNA, moderate soil drying, inferior spikelets, rice

## Abstract

Poor grain filling of inferior spikelets, especially in some large-panicle rice varieties, is becoming a major limitation in breaking the ceiling of rice production. In our previous studies, we proved that post-anthesis moderate soil drying (MD) was an effective way to promote starch synthesis and inferior grain filling. As one of the most important regulatory processes in response to environmental cues and at different developmental stages, the function of alternative splicing (AS) has not yet been revealed in regulating grain filling under MD conditions. In this study, AS events at the most active grain-filling stage were identified in inferior spikelets under well-watered control (CK) and MD treatments. Of 16,089 AS events, 1840 AS events involving 1392 genes occurred differentially between the CK and MD treatments, many of which function on spliceosome, ncRNA metabolic process, starch, and sucrose metabolism, and other functions. Some of the splicing factors and starch synthesis-related genes, such as SR protein, hnRNP protein, *Os*AG*PL2*, *OsAPS2*, *OsSSIVa, OsSSIVb*, *OsGBSSII*, and *OsISA1* showed differential AS changes under MD treatment. The expression of miR439f and miR444b was reduced due to an AS event which occurred in the intron where miRNAs were located in the MD-treated inferior spikelets. On the contrary, *Os*AG*PL2*, an AGPase encoding gene, was alternatively spliced, resulting in different transcripts with or without the miR393b binding site, suggesting a potential mechanism for miRNA-mediated gene regulation on grain filling of inferior spikelets in response to MD treatment. This study provides some new insights into the function of AS on the MD-promoted grain filling of inferior spikelets, and potential application in agriculture to increase rice yields by genetic approaches.

## 1. Introduction

Rice (*Oryza sativa* L.) is a staple food for more than half of the world’s population. However, its current levels of production cannot meet the demand driven by rapid population growth and economic development [1]. Hence, there is an urgent need to improve global rice production. China officially launched the Super Rice Breeding Program in 1996, aiming to cultivate new rice varieties with a high yield [2]. However, most of the super rice cultivars failed to achieve the high yield, mainly due to the poor grain filling of the later-flowering inferior spikelets [3,4].

The inferior spikelets flower later, exhibit a slower rate of increase in dry weight during grain development, and have a lower grain weight [4]. Many measures, such as moderate soil drying (MD) irrigation [5,6,7], rising CO_2_ concentration, and applications of plant hormones [8,9,10] have been used to improve the grain filling of inferior grains. The main component of rice endosperm is starch, and defects in starch synthesis will usually lead to incomplete grain filling. Four classes of enzymes, including ADP-glucose pyrophosphorylase (AGP), starch synthase (SS), starch branching enzyme (SBE), and starch debranching enzyme (DBE) are the most important enzymes in rice grain filling, which determine the grain weight and starch quality [11,12]. Numerous sources have shown that the manipulation of the starch biosynthesis pathway by modern molecular genetic techniques will alter the grain filling. It is also well demonstrated that grain filling is sensitive to different environmental cues [13,14], of which MD treatment has been proven to be a highly effective means of increasing inferior grain filling and grain yield by improving the key enzymes in sucrose-to-starch conversion [5,6,15]. The enzyme activities, especially sucrose synthase (SUS) and AGP in the inferior spikelets, are significantly enhanced by MD treatment, resulting in am improved inferior grain filling rate, grain weight, and yield [7,16]. Antagonism or synergistic interaction between ABA, ethylene, GA, and IAA play a critical role during grain filling under MD treatment [5,7,9]. Although it has been demonstrated that MD treatment regulated the hormonal balance, which thus played facilitatory roles during the grain filling in the inferior spikelets, the critical mechanism underlying the regulation of starch biosynthesis has yet to be well established.

Alternative splicing (AS), a process that generates multiple distinct transcripts from a single multi-exon gene, is prevalent in plants and responds to environmental changes and stress treatments [17,18], including ultraviolet (UV) irradiation [19], temperature stress (cold and heat) [20,21], and cadmium stress [22]. As an important factor in gene regulation, AS is an emerging research area related to post-transcriptional regulation [23], and significantly affects crop growth and development. For instance, the pre-mRNA splicing of the *OsFCA* gene controls the developmental switch from the vegetative to the reproductive phase in Arabidopsis [24]. In addition, the *Waxy* gene encodes a granule-bound starch synthase that is necessary for the synthesis of amylose in endosperm. Alternative splicing, caused by a single mutation in the 5′ splice site of *Waxy*, results in a reduced level of amylose [25,26]. Most recently, AS of *OsGS1;1* affected grain filling by regulating the amylose content and sugar metabolism [27]. These studies indicate that AS affects plant development and is also involved in regulating grain filling.

MicroRNAs (miRNAs), another post-transcriptional regulation mechanism, can be classified as either intergenic miRNAs or intronic miRNAs [24]. In animals, pre-mRNA splicing has been shown to participate in both intergenic and intronic miRNA processing [28,29]. This mechanism has also been conducted in plant miRNA primary transcripts [30,31]. MiRNAs play roles usually through regulating their target genes [32], which would be interrupted by the AS-induced disruption of miRNA binding sites [33]. It was shown that the additional regulation conferred by alternative splicing may link spliceosome activity to the regulation of certain miRNA–target interactions [33]. Recent works suggest that miRNAs, including miR1861, miR1432, and miR397, contribute to grain filling by regulating the starch synthesis and phytohormone biosynthesis in response to MD treatment [34,35,36,37]. However, the function of AS and its interaction with the miRNAs induced by MD treatment in regulating grain filling has not been reported. The aim of the present investigation was to identify AS events and the regulatory mechanism of grain filling in rice inferior spikelets under MD treatment at the most active stage of grain filling. We expect to propose a mechanism of MD-induced AS and its interaction with miRNAs in regulating the grain filling of inferior spikelets.

## 2. Results

### 2.1. AS Events of Rice Inferior Spikelets in Response to Moderate Soil Drying during Grain Filling

RNA-seq and small-RNA analysis at the most active grain filling stage (9 days after anthesis (DAA)) revealed that both starch synthesis and phytohormone biosynthesis are both regulated directly by MD treatment and indirectly regulated through differentially expressed miRNAs in inferior spikelets in response to MD treatment [7,34]. Another significant advantage of RNA-seq analysis is that differences in splicing can be detected from the sequences of the various transcripts. In this study, rMATS was utilized to identify the frequency of the different classes of differential splicing in rice inferior spikelets in response to MD treatment during grain filling.

There are several AS mechanisms, including skipped exon (SE), alternative 5′ splice site (A5SS), alternative 3′ splice site (A3SS), mutually exclusive exon (MXE), and retained intron (RI) (Appendix A). Each of these AS events can result in distinct transcripts, and hence diverse biological functions. The AS events of the rice inferior spikelets were analyzed using the rMATS software (http://rnaseq-mats.sourceforge.net/index.html; Version 4.1.0, accessed on 11 July 2022). As shown in Figure 1A and Appendix A, large numbers of AS events (CK, 14,264; MD, 15,788) were detected in the rice inferior spikelets during grain filling. Intriguingly, all of the AS types, including SE, A5SS, A3SS, MXE and RI, were significantly increased under MD treatment. Moreover, SE represented the largest proportion of AS events, at 45.57%. The percentages of MXE, A5SS, RI, and A3SS over the total AS event types were 2.42%, 12.38%, 17.44%, and 22.18%, respectively (Figure 1B).

### 2.2. MD-Induced AS Might Be Involved in Regulating Grain Filling of Rice Inferior Spikelets

On detailed analysis of the AS events, we found that the known AS events were the major event of rice inferior spikelets, accounting for 85.89% of the total AS events (Figure 2A). A total of 1840 differentially alternative splicing (DAS) events involving 1392 genes between the CK and MD treatments in inferior grains were obtained (Figure 2B). Compared with all of the AS types (Figure 1B), the percentages of A3SS and A5SS of DAS types increased by 8.25–9.14%, while MXE, RI, and SE had a 1.31–14.40% reduction (Figure 2C), suggesting that the AS events in inferior spikelets showed various susceptibilities to MD treatment.

Gene ontology (GO) and the Kyoto Encyclopedia of Genes and Genomes (KEGG) enrichment analysis (Figure 2D,E) were performed to identify the potential functions of these DAS genes. Of these, based on the GO database, the GO terms in the DAS genes were enriched in functions of biological process, including “Branched-chain amino acid metabolic process”, “Hexose metabolic process”, “Starch metabolic process”, “ncRNA metabolic process”, “Monosaccharide metabolic process”, “Starch biosynthetic process” and “RNA processing” (Figure 2D). The KEGG analysis of the DAS genes revealed that most of the genes were enriched in functions of “metabolism” and “genetic information processing”. “Other glycan degradation”, “RNA transport”, “Spliceosome” and “Starch and sucrose metabolism” were the most enriched KEGG pathways (Figure 2E). We then investigated the DAS genes related to the spliceosome pathway (Appendix A), which showed that the parts of the splicing factor encoding genes and the RNA recognition motif containing proteins were affected in the MD-treated grains (Appendix A), such as serine/arginine-rich (SR) protein subfamily gene (LOC_Os07g47630), RS domain with zinc knuckle protein (RSZ) subfamily gene (LOC_Os02g54770), plant-specific SC35-like splicing factor (SCL) (LOC_Os12g38430), heterogeneous nuclear ribonucleoprotein particle (hnRNP) protein gene (LOC_Os02g12850), and pre-mRNA-splicing factor SF2 (LOC_Os01g21420) (Figure 3). All of those genes may greatly alter the protein isoforms in comparison with the control group, and function as essential factors for constitutive and alternative splicing [38,39,40], which might in turn explain the increase in the AS events under MD treatment (Figure 1).

Given the GO and KEGG analysis results and the essential role of starch metabolism in grain filling, we focused on the DAS events of the starch synthesis-related genes and found that several DAS genes involved in starch synthesis had several AS forms under CK and MD treatments (Figure 4; Appendix A). ADP-glucose pyrophosphorylase genes (*Os*AG*PL2*, LOC_Os01g44220; *OsAPS2*, LOC_Os08g25734), key genes in regulating starch synthesis and grain filling [41] were reported that allosteric regulation on translated proteins of those genes has altered the catalytic activity of the cytoplasmic AGPase and starch biosynthesis [42]. In the present study, the several AS forms of the AGPase genes were identified under CK and MD treatments, including the known and novel AS events (Figure 4; Appendix A). Furthermore, several variants of other starch synthase-related gene transcripts, including *Os*AG*PL2*, *OsAPS2*, starch branching enzyme gene (*OsSBEI*, LOC_Os06g51084), granule-bound starch synthase gene (*OsGBSSII*, LOC_Os07g22930), soluble starch synthase genes (*OsSSIVa*, LOC_Os01g52250; *OsSSIVb*, LOC_Os05g45720), phosphoglucose isomerase gene (*OsPgi*, LOC_Os08g37380), isoamylase gene (*OsISA1*, LOC_Os08g40930; *OsISA3*, LOC_Os09g29404) and glucan phosphatase gene (*OsSEX4*, LOC_Os03g01750). These results indicated that the promotion of starch biosynthesis by MD treatment in inferior spikelets was partially mediated by the MD-induced AS during grain filling.

### 2.3. MD-Induced AS Event Influenced Primary miRNA Expression

The biogenesis of miRNAs relies on the coupled interaction of Pol-II-mediated pre-mRNA transcription and intron excision [43], during which the accurate splicing of the intron is critical for the efficient processing of the mRNA [21]. For this reason, we identified all of the RI-type AS events and the related intronic miRNAs in the inferior spikelets of rice under MD treatments. Hundreds of the RI-type AS events are organized in each chromosome, four of which identified a significant association between the AS events and intronic miRNAs (Figure 5). MiR439f, miR1847, miR444b, and miR1867 are located within the intronic regions of LOC_Os01g35930, LOC_Os01g36640, LOC_Os02g36924, and LOC_Os03g53190, respectively (Figure 5). The expression of miR439f and miR444b were reduced in MD-treated inferior spikelets, detected by small RNA-seq in our previous study (Appendix A) [34]. Those intronic miRNAs putative targets were predicted in previous studies [34,44], including MADS-box family gene (MADS) genes, AP2-like ethylene-responsive transcription factor genes, NAC domain-containing protein genes, and other transcription factors or genes. A granule-bound starch synthase gene (*OsGBSSII*) was found to be regulated by miR1867, as revealed by the 5′-RACE and degradome analysis [45]. These results indicate that AS acts as a regulatory mechanism for the miRNA processing in response to MD treatment, and might regulate grain filling via their target genes.

### 2.4. Identification of miRNA Binding Sites Disturbed by AS of Rice Inferior Spikelets under Moderate Soil Drying Post-Anthesis

Plant miRNAs recognize their target mRNAs through perfect or near perfect base pairing, which can be blocked by disrupting the miRNA binding sites (MBS) by AS [33,46]. *Os*AG*PL2*, a key gene in regulating starch synthesis, has seven different transcripts with a varying length of 5′ UTR (ranging from 218 bp to 600 bp) by AS (Figure 6). Based on the transcriptome results and AsmiR tools (http://forestry.fafu.edu.cn/bioinfor/db/ASmiR/; accessed on 1 April 2022), there was an AS region in the interval of 25,354,183–25,355,073 bp on Chr1 at the 5′ UTR of *Os*AG*PL2* that contains a functional binding site for miR393b. Several alternatively spliced transcripts of rice inferior spikelets do not contain the MBSs. Therefore, the miR393b-*Os*AG*PL2* interaction network in rice spikelets might be disrupted by AS on the binding sites under CK and MD treatments, which could be a potential novel mechanism of MD-promoted grain filling of inferior spikelets.

## 3. Discussion

### 3.1. DAS Events in Response to MD Treatment Participate in Starch Biosynthesis and Contribute to Increased Grain Filling in Inferior Spikelets

The AS of the pre-mRNAs from multiexon genes allows organisms to increase their coding potential and regulates the gene expression through multiple mechanisms. AS is involved in most of the plant processes, including plant growth, development, and responses to external cues [47]. The function of AS on grain filling and yield has been a research hotspot for many years [26,27,48,49,50,51,52]. A single mutation at the 5′splice site of *Waxy* affects the alternative splicing of its pre-mRNA, resulting in the reduced levels of amylose [25]. The AS of *OsbZIP58* may contribute to heat tolerance, and have an effect on some of the starch-hydrolyzing α-amylase genes during grain filling [52]. Alternative splicing of *OsLG3b* has also been reported to control grain length and yield in rice [50]. When*TaGS3* undergoes AS, it produces five splicing variants, resulting in opposite effects on grain weight and grain size [51]. Recently, a study also demonstrated that the AS of *GS1;1* affects grain amylose content and sugar metabolism in rice [27]. In the present study, a total of 1840 DAS events, including *OsbZIP58*, *OsLG3b*, and *OsAGPL2*, were identified under CK and MD treatments in inferior spikelets (Appendix A), suggesting that the AS events in response to MD treatment also participate in starch biosynthesis and contribute to increased grain yield.

Furthermore, by analyzing the GO and KEGG signaling pathways of the DAS genes, starch biosynthetic and sucrose metabolism were also represented in the most enriched KEGG pathways (Figure 2D,E), including *OsAGPL2*, *OsAPS2*, *OsSSIVa*, *OsSSIVb*, *OsGBSSII*, *OsISA1*, etc. In rice seed endosperm, the cytosolic AGP isoform, the OsAGPS2b/OsAGPL2 complex, catalyzes the limiting step and plays a key role in starch synthesis [41,53]. SSIV is considered important for the initiation of starch granules [54]. *OsISA1* is one of the most important genes determining the starch structures in rice grains, which is directly involved in the synthesis of amylopectin [55]. These results indicate that the MD promotion of starch synthase and biosynthesis was mediated partially by the MD-induced AS during grain filling.

### 3.2. AS of Pre-mRNAs of Splicing Factors Increases the Complexity of Inferior Grain Filling Response to MD Treatment

SR proteins and hnRNP proteins are the main families of splicing factors, which guide spliceosomal components and thereby the spliceosome to the respective splice sites [47,56,57]. They are essential splicing factors required for both constitutive and alternative splicing [17]. Several reports indicate that various biotic and abiotic stresses influence the AS of pre-mRNAs of many spliceosomal proteins. The SR proteins, in particular, undergo extensive alternate splicing [17,18,20]. Accumulating evidence suggests that manipulating SR protein expression subsequently alters the splicing of other pre-mRNAs, including SR pre-mRNAs [17,57]. In this study, the spliceosome pathway was also one of the most enriched pathways in the inferior spikelet under MD conditions (Figure 2E), leading to an increase in the AS events. The spliceosome-related genes undergo AS to produce multiple transcripts under MD treatments, including SR protein subfamily genes, RSZ subfamily genes, plant-specific SCL subfamily genes, and other splicing factors genes (Figure 3). This result may also in turn explain the increase in the AS events under MD treatment (Figure 1). Thus, the MD-induced AS of pre-mRNAs of splicing regulators increases the complexity of gene regulation, which may also contribute to the increased grain filling in the inferior spikelets.

### 3.3. MD-Induced AS Provides a Mechanism for the Regulation of miRNAProcessing, Leading to Increased Grain Filling in Inferior Spikelets

The AS of pre-mRNAs are widespread in eukaryotes, and generate different mature RNA isoforms from the same primary transcript, ensuring the proper expression of the genome and the higher proteome diversity [33]. Several intronic miRNAs have been discovered in plants [58,59], some of which have potential AS isoforms that may be affected by the AS events triggered under specific conditions [58,60]. The miR400 case nicely illustrates this issue, providing direct evidence that AS acts as a regulatory mechanism for miRNA processing [21]. In this study, we identified all of the RI-type AS events containing miRNAs. Four relating events particularly attracted our attention (Figure 5). MiR439f, miR1847, miR444b, and miR1867 are located within the intronic regions of its host, whose expressions were reduced in MD-treated inferior spikelets (Appendix A), indicating that intronic splicing may regulate the miRNA expression in response to MD treatment. Recently, growing evidence has demonstrated that the miRNAs also play crucial roles in controlling grain filling [34,35,44,61]. As reported previously, miR1867 was highly expressed during grain filling, and can regulate the genes implicated in the starch synthesis pathways in rice [45]. Therefore, we conclude that the MD-induced AS provides a possible mechanism for the regulation of microRNA processing, which may also contribute to the increased grain filling in inferior spikelets.

### 3.4. AS of Target Genes Increases the Complexity of miRNAs Regulation of Starch Biosynthesis

As key post-transcriptional regulators, miRNAs regulate their target genes by binding to the complementary MBS in the target mRNAs. A higher frequency of AS at MBSs and alternatively spliced MBSs enhances the regulatory complexity of the miRNA-mediated gene networks [33]. For example, AS produces multiple *SPL4* mRNA isoforms, with or without the binding site for miR156, resulting in an accelerated rate of flowering induction and significantly fewer adult leaves [33]. Os01g31870.8, one of the shortest transcript variants of *OsNramp6*, were downregulated by miR7695, which was the only transcript containing MBSs complementary to miR7695 [62]. *OsAGPL2*, a AGPase gene, plays a pivotal role in starch biosynthesis in higher plants, whose activity is directly determined by the population of the allosteric regulation [42]. The genomic data indicated that seven transcript variants of *OsAGPL2* were produced by alternative splicing (Figure 6). Among the various *OsAGPL2* splice variants, only three transcript variants contained complementary sites for miR393b-derived small RNAs, which were located at the 5′ UTR region of those transcript variants. All of those results indicate that the AS of *OsAGPL2* may attenuate miR393b-mediated gene regulation and ultimately lead to altered levels of enzyme activity. Furthermore, the miR393-overexpressed transgenic lines have smaller seed size compared with the wild type [63]. The expression of miR393b-3p was also reduced by MD treatment in the inferior spikelets [34]. Thus, the MD-induced AS of MBS increases the complexity of the miR393b-*OsAGPL2* gene regulation and function on starch biosynthesis regulation.

## 4. Materials and Methods

### 4.1. Plant Material and Experimental Design

The plant materials and growth conditions were described in detail in our previous article [7]. Briefly, the experiment was conducted in a greenhouse in the normal rice growing season in Changsha, China. Nipponbare (Nip) was used in this study. The seedings were raised at a hill spacing of 0.2 m× 0.2 m and two seedings per hill. Fertilizer and pesticide treatments were applied, according to normal agricultural practices, as described previously by Zhang et al. [16]. All of the plants were maintained well-watered up to 6 DAA. The soil water potential was monitored in the MD treatment with two tensiometers (Institute of Soil Science, Chinese Academy of Sciences, Nanjing, China) installed at a depth of 30 cm. The pots were not irrigated until the soil water potential reached −25 kPa. The CK plants were used as a blank control. The inferior kernels from each treatment were sampled at 9 DAA with three biological replicates, and preserved in a refrigerator at −80 °C for RNA sequencing.

### 4.2. RNA-seq Analysis

The RNA sequencing and analysis were performed by Majorbio Bio-pharm Technology Co., Ltd. (Shanghai, China). The data were analyzed on the online platform Majorbio Cloud Platform (www.majorbio.com; accessed on 1 November 2020). Briefly, the total RNA was extracted using Plant RNA Purification Reagent, according to the manufacturer’s instructions (Invitrogen, Carlsbad, CA, USA). The construction of the sequencing libraries, analysis, qualification, and paired-end RNA-seq sequencing were described by Teng et al. [7]. The transcriptomic data were submitted to the NCBI, and the relevant accession number is PRJNA728244.

### 4.3. Alternative Splicing Analysis

The rMATS software with default settings was used to detect the differential alternative splicing events from RNA-seq data [64]. The online platform PlantSPEAD (http://chemyang.ccnu.edu.cn/ccb/database/PlantSPEAD; accessed on 1 November 2020) was used to perform the splicing factors’ analyses [65]. The mRNA-Seq alignment files generated by HISAT2 (http://ccb.jhu.edu/software/hisat2/index.shtml; accessed on 1 November 2020) were used as an input for the rMATS analysis. The rice genome annotation project (http://rice.plantbiology.msu.edu/; accessed on 1 November 2020) was used as a reference, with default parameter settings. Finally, the rMATS was used to calculate the *p*-value for the AS events among the different treatments. The DAS events were extracted with a *p*-value ≤ 0.05. Splicing models of some of the DAS events were exported from the rMATS software and processed. To visualize the rMATS results and splicing model of DAS events, Majorbio Cloud Platform was used.

### 4.4. GO and KEGG Pathway Analyses

In addition, functional enrichment analysis, including GO and KEGG was performed to identify which DAS genes were the top 20 most enriched in GO (http://www.geneontology.org/; accessed on 1 April 2022) and KEGG (http://www.genome.jp/kegg/; accessed on 1 April 2022) database, compared with the whole-transcriptome background.

## 5. Conclusions

Taken together, the RNA-sequencing and small RNA-sequencing were used to reveal the interaction between AS and miRNAs in inferior spikelets under moderate soil drying. The results show that post-anthesis moderate soil drying differentially affected the AS of genes, many of which function on spliceosome and starch and sucrose metabolism. The MD-induced AS could be a potential mechanism modulating the expression of intronic miRNAs and their function on grain filling. Alternatively, the AS of MBSs is also a plausible mechanism for miRNA-mediated gene regulation on inferior grain filling under MD treatment. To summarize, this study expands our understanding of the AS function on the MD-promoted grain filling of inferior spikelets.

## Figures and Tables

**Figure 1 ijms-23-07770-f001:**
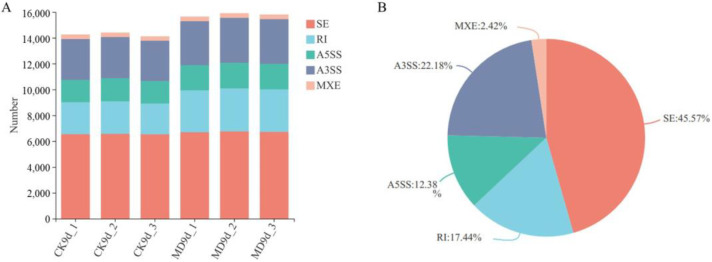
Summary of AS type of rice inferior spikelets under CK and MD conditions. (**A**) Number of the identified AS events. SE, skipped exon; A5SS, alternative 5′ splice site; A3SS, alternative 3′ splice site; MXE, mutually exclusive exon; RI, retained intron; (**B**) Summary of AS type, represented as percentages.

**Figure 2 ijms-23-07770-f002:**
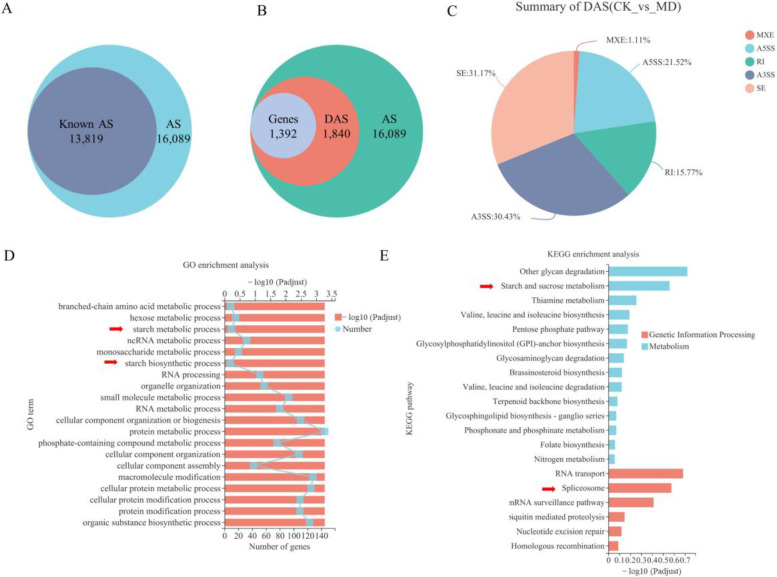
Identification of DAS events and most enriched GO and KEGG pathways of DAS genes in inferior kernels in the comparison of CK and MD at 9 DAA. (**A**) Proportion of known AS in all AS events; (**B**) Proportion of DAS in all AS events; (**C**) Summary of DAS type, represented as percentages; (**D**) The top 20 most enriched GO pathways of DAS genes; (**E**) The top 20 most enriched KEGG pathways of DAS genes. Several prominent signaling pathways were annotated with red arrow.

**Figure 3 ijms-23-07770-f003:**
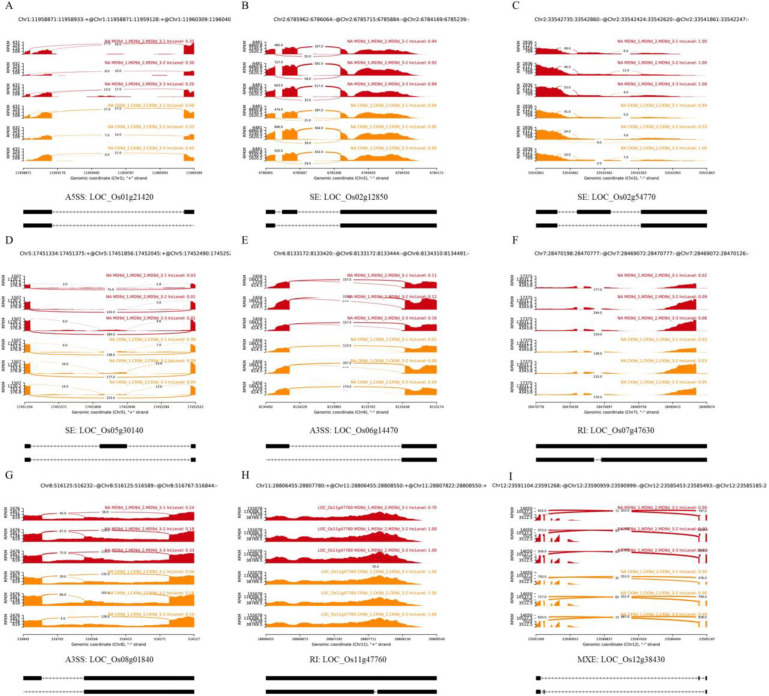
Quantitative visualization (Sashimi plot) of some DAS events of spliceosome pathway genes of rice inferior spikelets in the comparison of CK and MD at 9 DAA. (**A**) LOC_01g21420; (**B**) LOC_02g12850; (**C**) LOC_02g54770; (**D**) LOC_05g30140; (**E**) LOC_06g14470; (**F**) LOC_07g47630; (**G**) LOC_08g01840; (**H**) LOC_11g47760; (**I**) LOC_12g38430. Each track visualizes the splicing event within the biological replicates for the CK (orange) and MD (red) treatment samples. Count values on curved lines describe the coverage within the splice junction. The left scale presents the coverage depth in the range of the AS region. IncLevel are presented on the right side for each track. AS models are indicated below the figure.

**Figure 4 ijms-23-07770-f004:**
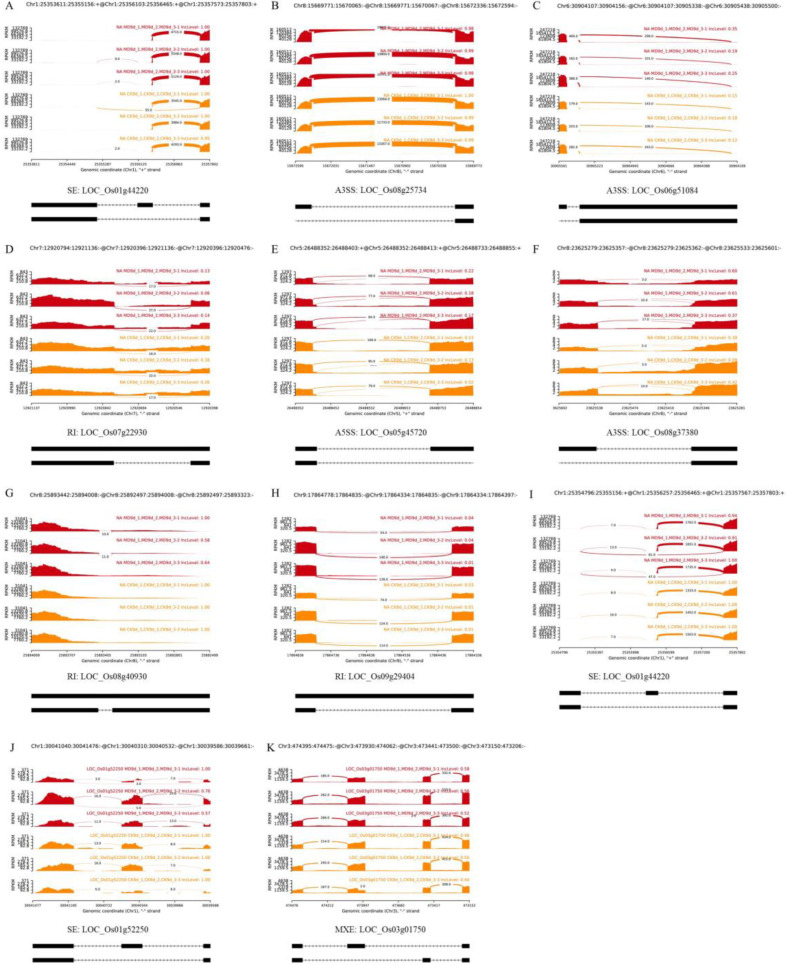
Quantitative visualization (Sashimi plot) of some DAS events of starch synthesis-related genes of rice inferior spikelets in the comparison of CK and MD at 9 DAA. Known AS events (**A**–**H**) and novel AS events (**I**–**K**) of starch synthesis-related genes in inferior kernels under CK and MD treatments. (**A**) LOC_01g44220; (**B**) LOC_08g25734; (**C**) LOC_06g51084; (**D**) LOC_07g22930; (**E**) LOC_05g45720; (**F**) LOC_08g37380; (**G**) LOC_08g40930; (**H**) LOC_09g29404; (**I**) LOC_01g44220; (**J**) LOC_01g52250; (**K**) LOC_03g01750.

**Figure 5 ijms-23-07770-f005:**
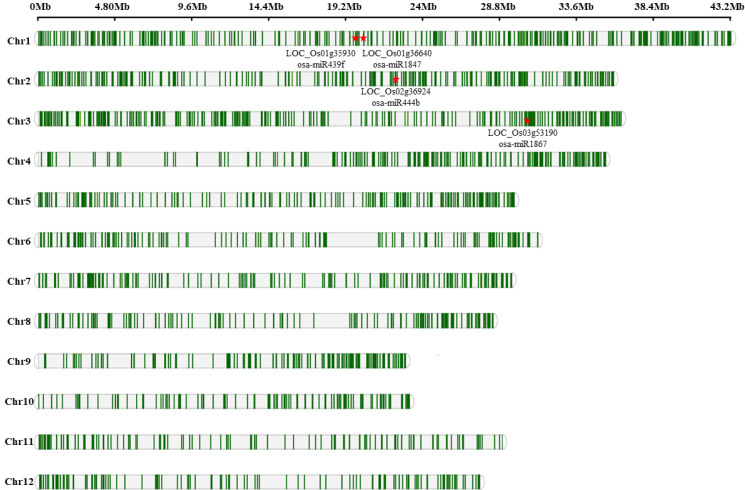
Distribution of RI-type AS events and the related intronic miRNAs in inferior spikelets of rice under CK and MD treatments. The location of the RI-type AS events with intronic miRNAs are indicated with red stars.

**Figure 6 ijms-23-07770-f006:**
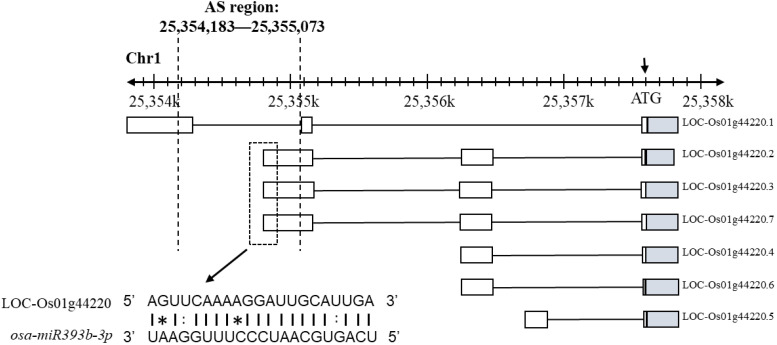
Alternative splicing of the miR393b binding site of *OsAGPL2* 5′ UTR of rice inferior spikelets under moderate soil drying post-anthesis. White boxes represent noncoding exons and shaded boxes represent coding exons. Dashes indicate Watson–Crick base pairing, colons indicate G-U base pairing, and asterisks indicate other non-canonical pairs.

## Data Availability

All of the data generated or analyzed during this study are included in this published article.

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
