# Peer review of "Moderate Soil Drying-Induced Alternative Splicing Provides a Potential Novel Approach for the Regulation of Grain Filling in Rice Inferior Spikelets"

_ijms, 2022, doi:10.3390/ijms23147770_

Round 1
Reviewer 1 Report
Comments regarding the paper “Moderate soil drying-induced alternative splicing provides a potential novel approach for the regulation of grain filling in rice inferior spikelets”
This study assessed the influence of moderate soil drying on alternatively spliced gene products and miRNAs involved in the grain filling stage of rice development. I think this is an interesting study as a first step to exploring the molecular basis of crop development. However, I did have a few questions regarding the study. If the authors would address these questions, I think it would help the readers get through the manuscript more easily.
A number of genes are listed as candidates for the potential causes of the increased grain filling under MD conditions, so wouldn’t the logical next step be to try to transform some of these genes and see which, if any, are actually responsible for the increased grain filling under the MD conditions?
Instances or RI type AS events were looked into in greater depth at line 183, but why aren’t any of the other four AS types studied in equal depth?
You find a potential binding site for miR393b, but why wasn’t this also done for the four miRNAs discussed in the prior section, miR439f, miR1847, miR444b, and miR1867? The same can be said for all the genes mentioned that are involved in grain filling - did any of these have binding sites for the miRNA? Was that evaluated?
Lastly, I was able to access one of the supplementary files, which had multiple sheets that weren't labeled to correspond to any of their supplementary figures as far as I could tell, but the other file was corrupted.
Suggested grammatical edits:
L19g: Use “were” instead of “was”
L19: What does “CK” stand for? (I assume it’s an abbreviation for “check” or “control” but it needs to be explicitly stated as such).
L20g: Replace “were differentially occurred” with “occurred differentially”
L21g: Change it to “treatments, many of which…”
L22g: starch, and sucrose metabolism, and other functions.
L24g: OsGBSSII, and OsISA1
L24g: “showed”, “AS”, and “changes” shouldn’t be in italics
L25g: an AS event which occurred…
L29g: change “in” to “into”
L36g: change “demand for rapid…” to “demand driven by rapid…”
L40:g change “high yield mainly ascribed to…” to high yield mainly due to…”
L44g: CO2 concentrations, and applications of plant hormones have been…
L45g: used to improve the grain filling of inferior grains…
L46g: change “is starch, defects…” to “is starch, and defects…”
L50g: change “Numerous evidences shown…” to “Numerous sources have shown…”
L51g: Change “manipulations of starch biosynthesis pathway…” to “manipulation of the starch biosynthesis pathway…”
L53g: Change “environmental cues. Of which…” to “environmental cues, of which MD treatment has been proven to be a highly effective means of increasing inferior grain filling and grain yield…”
L56g: Change “spikelets are significantly…” to “spikelets, are significantly…”
L58g: Change “GA, IAA…” to “GA, and IAA…”
L60g: Change “effects” to “roles”
L64g: Change “responses” to “responds”
L68: What is FCA?
L69g: Add “The” before “Waxy gene…”
L69: Is the Waxy gene somehow related to the FCA pre-mRNA? If not, this is a confusing transition.
L70g: Replace “A single mutation…” with “Alternative splicing caused by a single mutation in the 5’ splice site of Waxy results in reduced levels of amylose.”
L 73g: Change “indicated” to “indicate”
L76: What’s the difference between intergenic and intronic miRNAs?
L80g: Change “blunted” to “interrupted”
L81g: “Showed” to “shown”
L83/84g: Move “in response to MD treatment” to the end of the sentence.
L85g: Make “microRNA” plural
L92g: Make “responses” “response”
L93: DAA=Days after anthesis?
L95g: Remove “or”, assuming I’m interpreting that sentence correctly
L98g: Change “splicing of rice” to splicing in rice”
L100g: Change “several forms of AS mechanisms” to “several AS mechanisms”
L101g: is MXE supposed to be mutual exclusion exon?
L101g: “(MXE), and retained intron…”
L102: not able to access figure S1
L103: Are some of these biological functions more relevant or important for the context of this paper?
L104: Unable to discern the supplemental information- the tables are not labeled
L110: On the y-axis, number of what? What are the conditions displayed on the x-axis, because they haven’t been explained at this point in the paper? There is no indication of significance in part A. Does part B related to CK or MD conditions?
L111: “Summary of AS type of rice inferior spikelets”- is this supposed to mean the type of splicing occurring within that tissue?
L116/117: This is unclear, AS events account for 85.89% of AS events? Something is wrong in this sentence. Also, figure 1A has no numerical values that would support this claim. Is this supposed to refer to figure 2?
L118: Where are these numbers derived from? What is meant by differentially alternative splicing and how is it determined?
L119: How does figure 2B convey any information about the 1392 genes mentioned?
L121g: Change “having” to “had”
L128g: In “enrichment GO terms in DAS genes were enriched…” get rid of “enrichment”
L137: Unable to access supplementary information, you probably only need to refer to the figure once in the sentence.
L139: What is “RSZ”?
L140: What is “SCL”?
L143g: Get rid of “of these proteins”
L145: What is meant by “the increment of the proportion”?
L150: I’m not really sure how to interpret Figure 2C, since it seems like it should be for either CK or MD conditions, but the comparison between the two is confusing.
L153: Increase the resolution of the images so the values can be read when zoomed in, or reduce the number of plots shown. Also, I’m not able to notice any appreciable differences between the control and MD tracks.
L154: The figure caption doesn’t mention anything about what the 9 different plots are, what are A-I showing?
L160g: Change it to “and the essential role of starch metabolism…”
L161g: “we focused on”
L163-165g: “In ADP… and key genes…, it was shown that allosteric regulation on translated proteins has altered…”
L166: AGPase or AGP?
L168g: This sentence needs to be restructured so it’s easier to read
L169-177: You list all the genes with alternate splice variants, but do all of these genes now have increased activity/expression as a result of the AS event, or are they just different and that’s what you’re attributing the increased grain filling to?
L178: Increase the resolution of the images
L184: “Intronic miRNAs are located within the intronic regions.” Does this really need to be said?
L186g: Change it to “during which the accurate splicing of introns is critical for efficiently processing the mRNA.”
L194: Cite the study you’re referring to
L195g: Change “study” to “studies”
L201: What about the targets of miR1847?
L213g: Change “a” to “an”, make “intervals” singular.
L219: Do the asterisks indicate the miRNA binding site?
L224g: Either use “DAS” or spell out the whole thing, not just “differentially”
L226: Why put the explanation for pre-mRNA when you’ve already used it so often?
L230: “The Waxy…” – wasn’t this already talked about in the results?
L237g: “Recently, a study…GS1;1 affects”
L240: Figure 2 shows the 1840, but doesn’t depict any of the individual genes you mention here
L244: Most? How is this quantified?
L246: None of the genes can be identified based on the information you provide in figures 3 and 4
L246/247g: This shouldn’t be italicized
L258g: “many spliceosomal proteins. RS proteins in particular undergo extensive alternate splicing.”
L260g: “expression subsequently altering the splicing…”
L261g: This needs to be restructured to make sense
L263g: This fragment seems like it was intended to be part of the previous sentence
L266g: Do you mean the changes of the proportions seen between the two conditions?
L272: You already made this point on line 124
L275g: “AS isoforms that may be affected by…”
L278g: Use “Four” rather than 4
L283g: Is it really crosstalk, or is it just the involvement of miRNA in grain filling?
L291g: “…frequency of AS at MBSs…”
L294: What are the phenotypes?
L296g: “…which was the only transcript containing MBSs complementary to miR7695.”
L297g: “OsAGPL2, an AGPase…”. Also, I’m unsure what this sentence is supposed to convey
L299g: Use “seven” instead of 7, remove “gene” after OsAGPL2, and “from “
L304: I’m unsure of what’s meant by “activities change”
L310-330: There is no indication as to the population size or the number of libraries, which doesn’t instill much confidence.
L320: Why are you introducing what CK stands for at literally the last instance that you use it in the paper?
L321: Why explain the acronym after you’ve used it three times already?
L325: Get rid of “of”
L336: Get rid of “the”
L341/342: Why were only some of the DAS events looked at, and how were they processed?
L353: Should it be “miRNAs and their function…”
Author Response
Response to Reviewer 1 Comments
Comments regarding the paper “Moderate soil drying-induced alternative splicing provides a potential novel approach for the regulation of grain filling in rice inferior spikelets”
This study assessed the influence of moderate soil drying on alternatively spliced gene products and miRNAs involved in the grain filling stage of rice development. I think this is an interesting study as a first step to exploring the molecular basis of crop development. However, I did have a few questions regarding the study. If the authors would address these questions, I think it would help the readers get through the manuscript more easily.
Response to comments:
Many thanks for the reviewer’s valuable comments. We have revised our manuscript carefully according to the reviewer’s suggestions.
A number of genes are listed as candidates for the potential causes of the increased grain filling under MD conditions, so wouldn’t the logical next step be to try to transform some of these genes and see which, if any, are actually responsible for the increased grain filling under the MD conditions?
Response to comments:
Thanks for the comment. We agree with the reviewer. A number of AS events were differentially occurred between the CK and MD treatments. Many of which function on spliceosome and starch and sucrose metabolism. However, which gene is actually responsible for the increased grain filling under the MD conditions? We are also very interested in this question. So we have started to generate the mutant lines and the over expression plants of several candidate genes. We hope to provide some clues to this key question in our next article.
Instances or RI type AS events were looked into in greater depth at line 183, but why aren’t any of the other four AS types studied in equal depth?
Response to comments:
Thanks for the comment. MicroRNAs can be classified as either intergenic miRNAs or intronic miRNAs. Intronic miRNAs are located within the intronic regions. Biogenesis of miRNAs relies on the coupled interaction of Pol-II-mediated pre-mRNA transcription and intron excision, during which the splicing accuracy of intron is critical for the efficient processing. AS event may inhibited the miRNAs expression by holding the retained intron with the primary miRNA, but not by the alternative exon. Moreover, other four AS types do not impact on intron structure. For this reason, we identified all retained intron (RI) type AS events and the related intronic miRNAs.
You find a potential binding site for miR393b, but why wasn’t this also done for the four miRNAs discussed in the prior section, miR439f, miR1847, miR444b, and miR1867? The same can be said for all the genes mentioned that are involved in grain filling - did any of these have binding sites for the miRNA? Was that evaluated?
Response to comments:
Thanks for the comment. This is a good question. Indeed, we evaluated multiple miRNAs which are highly expressed in grains, including miR439f, miR1847, miR444b, and miR1867. miR393b was considered to be one the most possibly or probably related to grain filling, several alternatively spliced transcripts of which target gene do not contain the MBSs. Furthermore, miR393-overexpressed transgenic lines have smaller seed size compared with the wild-type(Bian et al.,2012).
Bian, H.; Xie, Y.; Guo, F.; Han, N.; Ma, S.; Zeng, Z.; Wang, J.; Yang, Y.; Zhu, M., Distinctive expression patterns and roles of the miRNA393/TIR1 homolog module in regulating flag leaf inclination and primary and crown root growth in rice (Oryza sativa). New Phytologist 2012, 196, (1), 149-161.
Lastly, I was able to access one of the supplementary files, which had multiple sheets that weren't labeled to correspond to any of their supplementary figures as far as I could tell, but the other file was corrupted.
Response to comments:
Thanks a lot for your comment. We are very sorry for this mistake, which has been corrected in the revised manuscript.
Suggested grammatical edits:
L19g: Use “were” instead of “was”
Response to comments:
Thank you for your suggestions, and we have corrected them in the new manuscript.
L19: What does “CK” stand for? (I assume it’s an abbreviation for “check” or “control” but it needs to be explicitly stated as such).
Response to comments:
Yes, we agree with the reviewer. Control has been used in this sentence.
L20g: Replace “were differentially occurred” with “occurred differentially”
L21g: Change it to “treatments, many of which…”
L22g: starch, and sucrose metabolism, and other functions.
L24g: OsGBSSII, and OsISA1
L24g: “showed”, “AS”, and “changes” shouldn’t be in italics
L25g: an AS event which occurred…
L29g: change “in” to “into”
L36g: change “demand for rapid…” to “demand driven by rapid…”
L40:g change “high yield mainly ascribed to…” to high yield mainly due to…”
L44g: CO2 concentrations, and applications of plant hormones have been…
L45g: used to improve the grain filling of inferior grains…
L46g: change “is starch, defects…” to “is starch, and defects…”
L50g: change “Numerous evidences shown…” to “Numerous sources have shown…”
L51g: Change “manipulations of starch biosynthesis pathway…” to “manipulation of the starch biosynthesis pathway…”
L53g: Change “environmental cues. Of which…” to “environmental cues, of which MD treatment has been proven to be a highly effective means of increasing inferior grain filling and grain yield…”
L56g: Change “spikelets are significantly…” to “spikelets, are significantly…”
L58g: Change “GA, IAA…” to “GA, and IAA…”
L60g: Change “effects” to “roles”
L64g: Change “responses” to “responds”
Response to comments:
Thanks for the suggestions. We are very sorry for this mistake, which has been corrected in the revised manuscript.
L68: What is FCA?
Response to comments:
We are very sorry for the mistake in the confused description. It should be OsFCA gene. We have corrected it in the new manuscript.
L69g: Add “The” before “Waxy gene…”
Response to comments:
Thanks for the suggestion, we have added “The” before “Waxy gene…” in the new manuscript.
L69: Is the Waxy gene somehow related to the FCA pre-mRNA? If not, this is a confusing transition.
Response to comments:
Sorry for the confused description. There was no relationship between FCA and Waxy gene. We have revised the manuscript thoroughly to eliminate the confusing statement.
L70g: Replace “A single mutation…” with “Alternative splicing caused by a single mutation in the 5’ splice site of Waxy results in reduced levels of amylose.”
L 73g: Change “indicated” to “indicate”
Response to comments:
Thanks for the suggestion. We have revised accordingly.
L76: What’s the difference between intergenic and intronic miRNAs?
Response to comments:
According to genomic organization, miRNA can be defined as intergenic miRNA and intronic miRNA. Intergenic miRNAs have independent transcription units (TUs), including promoter, transcript sequence and terminator units, and they do not overlap with other genes. However, intronic miRNAs are located within intronic regions of host genes.
L80g: Change “blunted” to “interrupted”
L81g: “Showed” to “shown”
L83/84g: Move “in response to MD treatment” to the end of the sentence.
L85g: Make “microRNA” plural
L92g: Make “responses” “response”
Response to comments:
Thanks for the suggestions. we have corrected them in the new manuscript.
L93: DAA=Days after anthesis?
Response to comments:
Yes, the abbreviation was defined in the new manuscript.
L95g: Remove “or”, assuming I’m interpreting that sentence correctly
Response to comments:
Sorry for the confused description. “or” should be retained. We want to say “both starch synthesis and phytohormone biosynthesis are regulated both directly by MD treatment and indirectly regulated through differentially expressed miRNAs in inferior spikelets in response to MD treatment”. We have revised the manuscript carefully to eliminate the confusing statement.
L98g: Change “splicing of rice” to splicing in rice”
L100g: Change “several forms of AS mechanisms” to “several AS mechanisms”
Response to comments:
Thanks for the suggestion. we have corrected them in the new manuscript.
L101g: is MXE supposed to be mutual exclusion exon?
Response to comments:
Sorry for the carelessness. We have revised it as “mutually exclusive exon” in the new manuscript.
L101g: “(MXE), and retained intron…”
Response to comments:
Thanks for the suggestion. we have corrected them in the new manuscript.
L102: not able to access figure S1
Response to comments:
Sorry for the carelessness. we have corrected them in the new manuscript.
L103: Are some of these biological functions more relevant or important for the context of this paper?
Response to comments:
Yes, we believe that grain filling in response to MD treatment is partly regulated by AS, which is also one of our major targets in this manuscript, especially the interaction between AS and microRNA in rice grains. However, there is still little direct experimental evidence for this function in rice.
L104: Unable to discern the supplemental information- the tables are not labeled
Response to comments:
Sorry for the carelessness. we have corrected them in the new manuscript.
L110: On the y-axis, number of what? What are the conditions displayed on the x-axis, because they haven’t been explained at this point in the paper? There is no indication of significance in part A. Does part B related to CK or MD conditions?
Response to comments:
Sorry for the confused description.
- y-axis indicates number of AS events detected in each of the samples. x-axis indicates samples.
- We have added the significance markers (asterisks) in the new supplementary table S1. However, there was no relation and significant difference between CK and MD conditions in proportion of AS events
- We have corrected them in the new manuscript.
L111: “Summary of AS type of rice inferior spikelets”- is this supposed to mean the type of splicing occurring within that tissue?
Response to comments:
Yes, it means the number and types of AS events of rice inferior spikelets under CK and MD conditions.
L116/117: This is unclear, AS events account for 85.89% of AS events? Something is wrong in this sentence. Also, figure 1A has no numerical values that would support this claim. Is this supposed to refer to figure 2?
Response to comments:
Sorry for the mistake. It should be " Figure 2A " instead of " Figure 1A " in line117. We have corrected them in the new manuscript.
L118: Where are these numbers derived from? What is meant by differentially alternative splicing and how is it determined?
Response to comments:
Thanks a lot for your comment. The numbers derived from Figure 2B. The rMATS software was used in this study to classify the AS events as five basic types and detect differential AS events from RNA-seq data. Differentially alternative splicing means there were some gene transcripts that significantly differed between the CK and MD conditions.
L119: How does figure 2B convey any information about the 1392 genes mentioned?
Response to comments:
Sorry for the carelessness. We have corrected them in the new Figure 2B.
L121g: Change “having” to “had”
L128g: In “enrichment GO terms in DAS genes were enriched…” get rid of “enrichment”
L137: Unable to access supplementary information, you probably only need to refer to the figure once in the sentence.
Response to comments:
Thanks for the suggestion. we have corrected them in the new manuscript.
L139: What is “RSZ”?
Response to comments:
Sorry for the carelessness. It should be “RS domain with zinc knuckle protein (RSZ) subfamily gene”, the abbreviation was defined in the new manuscript.
L140: What is “SCL”?
Response to comments:
Sorry for the carelessness. It should be “plant-specific SC35-like splicing factor (SCL)”, the abbreviation was defined in the new manuscript.
L143g: Get rid of “of these proteins”
Response to comments:
Thanks for the suggestion. we have corrected them in the new manuscript.
L145: What is meant by “the increment of the proportion”?
Response to comments:
Thanks a lot for your comment. Spliceosome was one of the most enriched KEGG pathways, which may resulted in more AS events and altering the protein isoforms. This can be a reason for the increment of AS event under MD treatment.
L150: I’m not really sure how to interpret Figure 2C, since it seems like it should be for either CK or MD conditions, but the comparison between the two is confusing.
Response to comments:
Thanks a lot for your comment. It means the summary of DAS type between CK and MD treatments, which was expressed as percentages.
L153: Increase the resolution of the images so the values can be read when zoomed in, or reduce the number of plots shown. Also, I’m not able to notice any appreciable differences between the control and MD tracks.
Response to comments:
Thanks for the suggestion. we have corrected them in the new manuscript.
L154: The figure caption doesn’t mention anything about what the 9 different plots are, what are A-I showing?
Response to comments:
Thanks a lot for your comment. Figure 3 showed the quantitative visualization of some spliceosome pathway genes. We want to point out that those genes had completely different AS forms under CK and MD treatments. The mean of A to I are shown in the new figure legend.
L160g: Change it to “and the essential role of starch metabolism…”
L161g: “we focused on”
L163-165g: “In ADP… and key genes…, it was shown that allosteric regulation on translated proteins has altered…”
Response to comments:
Thanks for the suggestion. we have corrected them in the new manuscript.
L166: AGPase or AGP?
Response to comments:
Thanks a lot for your comment. Here we use “AGPase” but not “AGP”.
L168g: This sentence needs to be restructured so it’s easier to read
Response to comments:
Thanks for the suggestion. we have corrected them in the new manuscript.
L169-177: You list all the genes with alternate splice variants, but do all of these genes now have increased activity/expression as a result of the AS event, or are they just different and that’s what you’re attributing the increased grain filling to?
Response to comments:
Thanks a lot for your comment. In the present study, several AS forms of starch synthase related genes were identified under CK and MD treatments. It has been shown that allosteric regulation on translated proteins of OsAGPL2 has altered the catalytic activity of the cytoplasmic AGPase and starch biosynthesis (Tuncel et al., 2014). In addition, MD has been well demonstrated to increase the enzyme activities for starch biosynthesis in many studies (Zhang et al., 2012, JXB; Wang et al., 2015 Planta; Teng et al., 2021, Plant J). We therefore speculate that promotion of starch biosynthesis by MD treatment was partially mediated by MD-induced AS during grain filling.
Tuncel, A.; Kawaguchi, J.; Ihara, Y.; Matsusaka, H.; Nishi, A.; Nakamura, T.; Kuhara, S.; Hirakawa, H.; Nakamura, Y.; Cakir, B., The rice endosperm ADP-glucose pyrophosphorylase large subunit is essential for optimal catalysis and allosteric regulation of the heterotetrameric enzyme. Plant and cell physiology 2014, 55, (6), 1169-1183.
L178: Increase the resolution of the images
Response to comments:
Thanks for the suggestion. we have corrected them in the new manuscript.
L184: “Intronic miRNAs are located within the intronic regions.” Does this really need to be said?
Response to comments:
The description has been deleted in the new manuscript.
L186g: Change it to “during which the accurate splicing of introns is critical for efficiently processing the mRNA.”
L194: Cite the study you’re referring to
L195g: Change “study” to “studies”
Response to comments:
Thanks for the suggestion. we have corrected them in the new manuscript.
L201: What about the targets of miR1847?
Response to comments:
Thanks a lot for your comment. Granule-bound starch synthase gene (OsGBSSII) was a direct target gene for miR1847, and plays a vital role in starch biosynthesis in grain filling, indicating regulation of miR1847 by AS event might eventually affect starch biosynthesis and grain filling
L213g: Change “a” to “an”, make “intervals” singular.
Response to comments:
Thanks for the suggestion. we have corrected them in the new manuscript.
L219: Do the asterisks indicate the miRNA binding site?
Response to comments:
Thanks a lot for your comment. Dashes indicate Watson-Crick base pairing, colons indicate G-U base pairing and asterisks indicate other non-canonical pairs. The explanation has been added in the new manuscript.
L224g: Either use “DAS” or spell out the whole thing, not just “differentially”
Response to comments:
Thanks for the suggestion. we have corrected them in the new manuscript.
L226: Why put the explanation for pre-mRNA when you’ve already used it so often?
L230: “The Waxy…” – wasn’t this already talked about in the results?
Response to comments:
Thanks a lot for your comment. The description has been deleted in the new manuscript.
L237g: “Recently, a study…GS1;1 affects”
Response to comments:
Thanks for the suggestion. we have corrected them in the new manuscript.
L240: Figure 2 shows the 1840, but doesn’t depict any of the individual genes you mention here
Response to comments:
Thanks a lot for your comment. The information of 1840 DAS events can be extracted from the additional file. “Figure 2” on line240 has been removed.
L244: Most? How is this quantified?
Response to comments:
Many thanks for the comment. We are very sorry for the mistake in the confused description. We have corrected them in the new manuscript to eliminate the confusing statement.
L246: None of the genes can be identified based on the information you provide in figures 3 and 4
Response to comments:
Many thanks for the comment. “Figure 3, 4” on line246 has been removed.
L246/247g: This shouldn’t be italicized
L258g: “many spliceosomal proteins. RS proteins in particular undergo extensive alternate splicing.”
L260g: “expression subsequently altering the splicing…”
L261g: This needs to be restructured to make sense
Response to comments:
Thanks for the suggestion. we have corrected them in the new manuscript.
L263g: This fragment seems like it was intended to be part of the previous sentence
Response to comments:
Yes. We have rephrased our descriptions in the new manuscript.
L266g: Do you mean the changes of the proportions seen between the two conditions?
Response to comments:
Yes.
L272: You already made this point on line 124
Response to comments:
Many thanks for the comment. The sentences on line 124 of the previous version of the manuscript have been removed from the revised version to avoid redundancy.
L275g: “AS isoforms that may be affected by…”
L278g: Use “Four” rather than 4
Response to comments:
Thanks for the suggestion. we have corrected them in the new manuscript.
L283g: Is it really crosstalk, or is it just the involvement of miRNA in grain filling?
Response to comments:
Thanks a lot for your comment. We are very sorry for this mistake, MiRNAs were proved to involve in controlling grain filling.
L291g: “…frequency of AS at MBSs…”
Response to comments:
Thanks for the suggestion. we have corrected them in the new manuscript.
L294: What are the phenotypes?
Response to comments:
Thanks a lot for your comment. “AS produces multiple SPL4 mRNA isoforms with or without the binding site for miR156, resulting in an accelerated rate of flowering induction and significantly fewer adult leaves”. we have corrected them in the new manuscript.
L296g: “…which was the only transcript containing MBSs complementary to miR7695.”
Response to comments:
Thanks for the suggestion. we have corrected them in the new manuscript.
L297g: “OsAGPL2, an AGPase…”. Also, I’m unsure what this sentence is supposed to convey
Response to comments:
Many thanks for the comment. This sentence has been removed from the revised version.
L299g: Use “seven” instead of 7, remove “gene” after OsAGPL2, and “from “
Response to comments:
Thanks for the suggestion. we have corrected them in the new manuscript.
L304: I’m unsure of what’s meant by “activities change”
Response to comments:
Sorry for the confusing description, and we have now rephrased our descriptions in the new manuscript.
L310-330: There is no indication as to the population size or the number of libraries, which doesn’t instill much confidence.
L320: Why are you introducing what CK stands for at literally the last instance that you use it in the paper?
L321: Why explain the acronym after you’ve used it three times already?
Response to comments:
Sorry for the confusing description and thank you for your suggestions. We have removed all the abbreviations (e.g., CK, MD, DAA) and rephrased our descriptions in the method section.
L325: Get rid of “of”
L336: Get rid of “the”
Response to comments:
Thanks for the suggestion. we have corrected them in the new manuscript.
L341/342: Why were only some of the DAS events looked at, and how were they processed?
Response to comments:
Thanks for the comment. Indeed, we evaluated all of the DAS events and only some of the DAS events most possibly related to grain filling were finally selected. The rMATS software with default settings was used to detect differential alternative splicing events from RNA-seq data (Shen et al.,2014). To visualize rMATS results and splicing model of DAS events, Majorbio Cloud Platform was used.
Shen, S.; Park, J. W.; Lu, Z.-x.; Lin, L.; Henry, M. D.; Wu, Y. N.; Zhou, Q.; Xing, Y., rMATS: robust and flexible detection of differential alternative splicing from replicate RNA-Seq data. Proceedings of the National Academy of Sciences 2014, 111, (51), E5593-E5601.
L353: Should it be “miRNAs and their function…”
Response to comments:
Thanks for the suggestion. we have corrected them in the new manuscript.
Reviewer 2 Report
Manuscript "Moderate soil drying-induced alternative splicing provides a potential novel approach for the regulation of grain filling in rice inferior spikelets" is very interesting.
Authors identified alternative splicing events and its regulatory mechanism of grain filling in rice inferior spikelets under moderate soil drying treatment at the most active stage of grain filling. Authors expected to propose a mechanism of moderate soil drying-induced alternative splicing and its interaction with miRNAs in regulating grain filling of inferior spikelets.
Material is described very good. Unfortunately, Authors not use any statistical analyses of data.
Paper needs major revision.
Author Response
Response to comments:
Many thanks for the comment. In this study, three biological repeats were used for RNA-sequencing. P-value ≤0.05 was used in sequencing data analysis, which is shown in supplemental tables. We have carefully revised our manuscript according to the constructive comments from all the reviewers. The structure of the article has been revised to provide more focus, and this is reflected in the new manuscript. We believe that our manuscript is substantially improved.
Reviewer 3 Report
I consider that the manuscript presented is of great value through the novelties that are presented here. While studying this manuscript I felt the need for a list of abbreviations. It is true that the necessary information can be found in the article, but are quite a lot and makes it difficult to read. Another observation would be that there is quite a bit of reference to previous studies, which is commendable, but I think there are certain explanations (such as experiment set-up) that could be improved. The data presented are overwhelmingly due to in silico analyzes, the presented data are quite thick and you have to be a real specialist in the field to understand what it is about, and if we consider the additional material, we can say that the study of this article is at least discouraging. That is why I consider that in the Conclusions section it could contain a more specific summary of what was done in this study.
Author Response
Response to comments:
We thank the reviewer for allowing us to organize our thoughts and data to make this easier for the reader to understand. In the revised version of the manuscript, we have modified the method and conclusions section according to all three reviewer’s comments. All the abbreviations were examined carefully and were explained when their appeared for the first time in the manuscript. Also, a specific summary of what has been done in this study was added in the conclusion section.
Round 2
Reviewer 2 Report
Now, all is ok.